# Sparsity-Based Joint Array Calibration and Ambiguity Resolving for Forward-Looking Multi-Channel SAR Imagery

**Jingyue Lu [1], Xuhua Wang [1,*], Yunhe Cao [2] and Lei Zhang [3]**

1   School of Computer Science and Technology, Xidian University, Xi'an 710071, China
2   National Laboratory of Radar Signal Processing, Xidian University, Xi'an 710071, China
3   School of Electronics and Communication Engineering, Sun Yat-sen University, Guangzhou 510275, China
*   Correspondence: wangxuhua@xidian.edu.cn

**Abstract:** Forward-looking multi-channel synthetic aperture radar (FLMC-SAR) can realize two-dimension image formation in monostatic mode. This system must face the problem of left–right Doppler ambiguity. In the traditional methods, the spatial degrees of freedom of the FLMC-SAR system is expected to achieve Doppler ambiguity resolving by beamforming approaches. However, the influence of array error on beamforming cannot be ignored. In practice, the array error will lead to the mismatch of the space–time characteristic, which will reduce the performance of the Doppler ambiguity resolving method based on beamforming. This paper proposes a sparsity-based joint array calibration and ambiguity resolving method to enhance the robustness of FLMC-SAR imagery. For the FLMC-SAR system, the space–time characteristic of targets is first analyzed, based on which the observation model of FLMC-SAR Doppler ambiguity combined with array error is derived. Then, the Doppler ambiguity resolving and array error estimation are transformed into a sparse recovery problem. A modified quasi-Newton method is proposed to realize the array error estimation and Doppler ambiguity resolving of all targets in the local area. Finally, the results of the simulation and the real-data experiments verify that the proposed method can achieve FLMC-SAR Doppler ambiguity resolving and imaging.

**Keywords:** forward-looking multi-channel synthetic aperture radar (FLMC-SAR); Doppler ambiguity; space–time characteristic; array calibration; sparse recovery

---

## 1. Introduction

Synthetic aperture radar (SAR) [1–7] is an important technique for achieving two-dimensional high-resolution microwave imaging. Due to the limitation of its working mechanism, SAR has a forward-looking blind area. By optimizing the system configuration of traditional SAR, SAR is used to obtain the two-dimensional image of the forward-looking area, called forward-looking SAR. The ability to acquire forward-looking two-dimensional images makes forward-looking SAR widely applied in practical engineering, such as aircraft blind landing, remote sensing, and so on. Forward-looking SAR systems can be divided into monostatic forward-looking SAR systems and bistatic forward-looking SAR systems.

Bistatic forward-looking SAR (BFSAR) [8–13] imaging is performed by equipping an additional receiver or transmitter platform to provide forward-looking Doppler diversity to achieve high cross-range resolution. The flexible system configuration makes BFSAR widely used. However, the system configuration requirements of BFSAR may not be met in some applications (such as having only one radar platform). Therefore, this paper focuses on the monostatic forward-looking SAR imaging system. Sector imaging radar for enhanced vision (SIREV) [14,15] is the earliest monostatic forward-looking SAR imaging system with an array of real apertures in the cross-trajectory direction. SIREV forms a synthetic aperture by controlling the timing of the transmitter and receiver to obtain the forward-looking two-dimensional images, which makes the synthetic aperture length

heavily dependent on the array aperture length. This means that the forward-looking resolution of the SIREV system is excessively dependent on the array aperture. However, a high-resolution forward-looking imaging requirement will cause higher system complexity, which limits the practical application of SIREV. Joint monopulse angle measurement [16] and SAR imaging is another forward-looking imaging technique. However, the accuracy of monopulse angle measurement for multiple targets in the same beam is limited, which limits the performance of the monopulse forward-looking SAR system for complex scenes. Multi-channel SAR [17–22] is used to obtain forward-looking images, known as forward-looking multi-channel SAR [23,24] (FLMC-SAR). However, Doppler ambiguity is a crucial problem that must be resolved in the imaging process. Beamforming is an excellent technique for FLMC-SAR to solve Doppler ambiguity. Unfortunately, in practical applications, the inevitable installation and other factors will introduce array errors, which is an essential factor that makes it impossible to achieve Doppler ambiguity resolving via beamforming. Rotating element electric field vector (REV) [25–27] is an effective method for array calibration. However, REV requires a set of calibration sources with known locations, which does not apply to FLMC-SAR imaging.

Sparsity-based algorithms [28–37] have shown great potential in radar imaging, such as image reconstruction from under-sampling rate image data [38–40], improving the resolution of the reconstructed image under the condition of constant sampling [41–45], and so on. At the same time, sparsity-based algorithms can also be used to estimate errors (such as motion and array) to improve imaging quality. The assumption of confirming the accuracy and success rate of image reconstruction is whether the reconstructed signal is sparse in the signal observation model, and the size of the observation matrix dimension also determines the computational complexity of the algorithm. Therefore, appropriate system resource allocation and reasonable imaging processes are key to ensure algorithm performance. In this paper, a sparsity-based joint array calibration and ambiguity resolving method is proposed to enhance the robustness of FLMC-SAR imagery. First, the space–time characteristic of FLMC-SAR is analyzed. The effect of array errors on Doppler ambiguity resolving indicates that the array errors will lead to the mismatch of space–time characteristics of the targets, causing performance degradation of Doppler ambiguity resolving. Considering the array errors, the observation model of space–time characteristic correction is derived. Then, the Doppler ambiguity resolving and array error estimation are transformed into a sparse recovery problem. Then, a modified quasi-Newton method is proposed to realize the array error estimation and Doppler ambiguity resolving in the local area. Finally, the results of the simulation and the real-data experiments verify that the proposed method can achieve FLMC-SAR Doppler ambiguity resolving and imaging. The main contribution can be summarized as follows.

(1) Both left–right Doppler ambiguity and array error estimation are considered in the observation model, which transforms the robust FLMC-SAR imaging without ambiguity into a sparse recovery problem. The modified quasi-Newton method is proposed to realizes both array error estimation and Doppler ambiguity resolving simultaneously.

(2) The sparse recovery problem is solved in the two-dimensional image domain after range–azimuth decoupling, which paves the way for sparse reconstruction in each range bin. The array error estimation and Doppler ambiguity resolving are realized in the local area, reducing the computational complexity of the proposed method. In addition, after pulse compression and azimuth focusing, the SNR of the two-dimensional image domain is significantly improved, which improves the image reconstruction accuracy.

The organizational structure of this article is as follows. In Section 2, the geometric model of FLMC-SAR is given. In Section 3, the observation model of FLMC-SAR is established. In Section 4, we propose the constrained optimization problem of array error estimation and introduce the joint Doppler ambiguity resolving and array error correction in detail. Section 5 gives the results based on simulation and real-data experiments. The results are discussed in Section 6. Section 7 summarizes the work of this paper.

## 2. FLMC-SAR Geometry Model

Figure 1 gives the FLMC-SAR geometry model. The Cartesian coordinate system is established. The origin $O$ is the projection of the aperture center on the ground, the trajectory direction is the $X$-axis, the array distribution direction is the $Y$-axis, and the vertical direction up to the ground is the $Z$-axis.

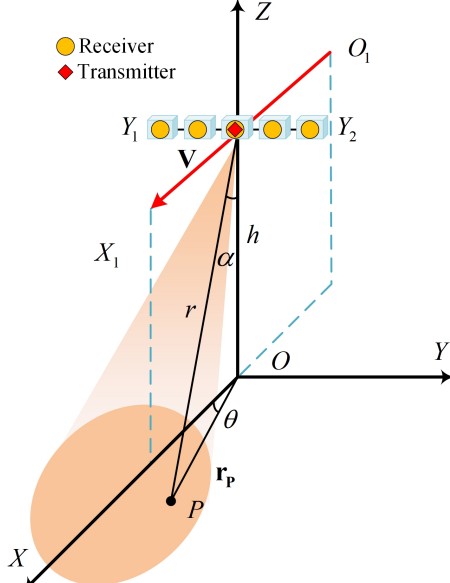

**Figure 1.** FLMC-SAR geometry model.

FLMC-SAR is a monopulse radar system. The radar platform moves along the trajectory $O_1 X_1$ at a speed $v$ and altitude $h$, which uses a single antenna as the transmitting antenna and a uniform linear array as the receiving antenna; $D$ denotes the length of the real antenna, and $d$ denotes the spacing between antenna elements. The receiver array antenna is symmetrically distributed along the $Y$-axis over $Y_1 Y_2$. The imaging area is located directly in the forward-looking area; $P$ denotes a target in the forward-looking imaging area. The radar's light of sight is defined as a vector pointing from the center of the aperture to the target. The radar's light of sight can be determined by the beam pointing pitch angle $\alpha$ between it and the $Z$-axis and the beam pointing azimuth angle $\theta$ between the projection of it on the ground and the $X$-axis. The reference slant range is defined as the distance between the target and the APC at the center of the aperture. With these parameters, the coordinates of the target $P$ in the $OXYZ$ coordinate system can be expressed as

$$\mathbf{P}(x, y, z) = (r \sin \alpha \cos \theta, r \sin \alpha \sin \theta, 0) \tag{1}$$

Let $t_m$ be the radar slow time; then the coordinate of the transmitter and receiver can be expressed as

$$\begin{cases} \mathbf{T_X}(vt_m, 0, h) \\ \mathbf{R_{Xi}}(vt_m, y, h) \end{cases} \tag{2}$$

where $y$ denotes the $Y$-coordinate of the receiver array element. Then the instantaneous two-way slant range of the target $P$ can be expressed as

$$R(t_m; r, \theta, \alpha, y) = R_R(t_m; r, \theta, \alpha, y) + R_T(t_m; r, \theta, \alpha) \tag{3}$$

where $R_T(t_m; r, \theta, \alpha)$ and $R_R(t_m; r, \theta, \alpha, y)$ denote the instantaneous slant range of the receiver and the transmitter, respectively.

$$R_R(t_m; r, \theta, \alpha, y) = \sqrt{(vt_m - r \sin \alpha \cos \theta)^2 + (y - r \sin \alpha \sin \theta)^2 + (r \cos \alpha)^2} \tag{4}$$

$$R_T(t_m; r, \theta, \alpha) = R_R(t_m; r, \theta, \alpha, 0) = \sqrt{(vt_m - r\sin\alpha\cos\theta)^2 + (r\sin\alpha\sin\theta)^2 + (r\cos\alpha)^2} \tag{5}$$

### 3. FLMC-SAR Observation Model

*3.1. FLMC-SAR Signal Model*

The radar system uses a linear frequency modulation (LFM) pulse signal, which is one of the most widely used signals in SAR system transmissions.

$$s(\tau) = \text{rect}\left(\frac{\tau}{T_p}\right) \cdot \exp\left[j2\pi\left(f_c\tau + \frac{\gamma}{2}\tau^2\right)\right] \tag{6}$$

where rect[] is the rectangular window function, $\tau$ is the radar fast time, $T_p$ is the time width of the LFM pulse, $f_c$ is the carrier frequency of the radar system, and $\gamma$ is the chirp rate of LFM signal.

Through down-conversion, the baseband received signal can be expressed as

$$
\begin{aligned}
s_P(\hat{t}, t_m; r, \theta, \alpha, y) = & A_P w_r\left[\hat{t} - \frac{R(t_m; r, \theta, \alpha, y)}{c}\right] \cdot \exp\left\{j\pi\gamma\left[\hat{t} - \frac{R(t_m; r, \theta, \alpha, y)}{c}\right]^2\right\} \\
& \cdot \exp\left[-j\frac{2\pi}{\lambda}R(t_m; r, \theta, \alpha, y)\right]
\end{aligned}
\tag{7}
$$

where $A_p$ is the reflectivity coefficient of the target $P$, $w_r$ is the envelope in range bin, and $c$ is the light speed.

Through pulse compression and range cell migration correction (RCMC), the signal can be expressed as

$$s_2(\hat{t}, t_m; r, \theta, \alpha, y) = A_P\text{sinc}\left\{B\left[\hat{t} - \frac{(r - r_0)}{c}\right]\right\} \cdot \exp\left\{-j\frac{2\pi}{\lambda}[R(t_m; r, \theta, \alpha, y) - R(t_m; r, 0, \alpha, 0)]\right\} \tag{8}$$

where $r_0$ is the reference slant range of the center of the imaging area.

It can be seen from Equation (8) that the signal of target $P$ has been focused on the range unit corresponding to the reference slant range $r$, so range–azimuth decoupling is achieved.

*3.2. Doppler Ambiguity in FLMC-SAR System*

To clarify the Doppler ambiguity, we rewrite Equation (8) in the following form:

$$
\begin{aligned}
s_2(\hat{t}, t_m; r, \theta, \alpha, y) = & A_P\text{sinc}\left\{B\left[\hat{t} - \frac{(r - r_0)}{c}\right]\right\} \cdot \exp\left\{-j\frac{2\pi}{\lambda}[\Delta R_1(t_m; r, \theta, \alpha, y)]\right\} \\
& \cdot \exp\left\{-j\frac{2\pi}{\lambda}[\Delta R_2(t_m; r, \theta, \alpha, y)]\right\}
\end{aligned}
\tag{9}
$$

$$\Delta R_1(t_m; r, \theta, \alpha, y) = R(t_m; r, \theta, \alpha, y) - R(t_m; r, \theta, \alpha, 0) \tag{10}$$

$$\Delta R_2(t_m; r, \theta, \alpha) = R(t_m; r, \theta, \alpha, 0) - R(t_m; r, 0, \alpha, 0) \tag{11}$$

where $\Delta R_1$ is the wave path difference between different channels caused by the different array element coordinate $y$, and $\Delta R_2$ is the slant range history of the target, which is the same between the different channels. As the array coordinate $y$ is much smaller than the reference slant range $r$, the far-field condition is satisfied. Equation (10) can be rewritten as

$$\Delta R_1(t_m; r, \theta, \alpha, y) \approx y\sin\alpha\sin\theta \tag{12}$$

Similarly, Equation (11) can be approximated as follows:

$$\Delta R_2(t_m; r, \theta, \alpha) \approx A + Bt_m \tag{13}$$

$$B = 2v \sin \alpha (1 - \cos \theta) \tag{14}$$

Then the Doppler frequency of the target can be expressed as

$$f_d(\theta, \alpha) = \frac{B}{\lambda} = \frac{2v \sin \alpha (1 - \cos \theta)}{\lambda}. \tag{15}$$

It can be seen from Equation (15) that the Doppler frequency of the target is a function related to azimuth angle $\theta$ and pitch angle $\alpha$, and the Doppler frequency is an even function of azimuth angle $\theta$. It means that targets with azimuth angles that are negative to each other have the same Doppler frequency. Traditional SAR imaging methods cannot distinguish the target with opposite azimuth angle in the same range unit, resulting in Doppler ambiguity.

As shown in Figure 2, in the imaging area, there are two targets $P_1$ and $P_2$ with the same reference slant range $r$, but opposite azimuth angles. In the ground grid, their coordinates can be expressed as

$$\begin{cases} P_1(r, \theta) \\ P_2(r, -\theta) \end{cases} \tag{16}$$

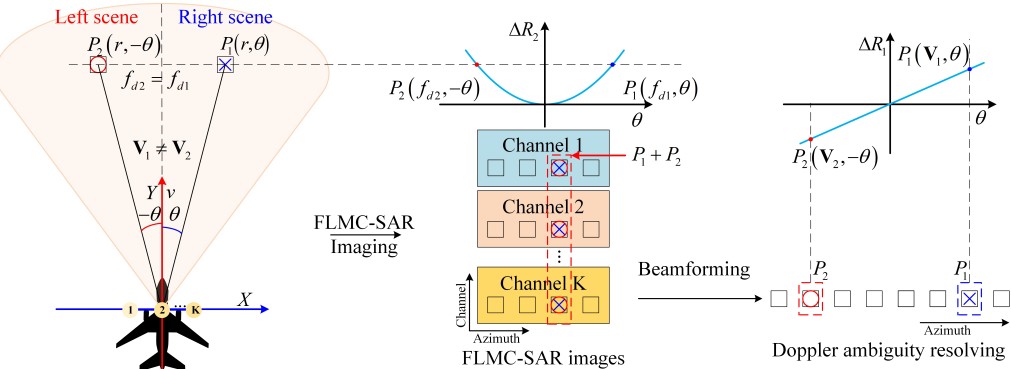

**Figure 2.** Doppler ambiguity in FLMC-SAR.

According to Equation (15), we can conclude that this Doppler frequency is the same. In the plane of the space–time spectrum $f_d - \theta$, there coordinates can be expressed as

$$\begin{cases} P_1(f_{d1}, \theta) \\ P_2(f_{d2}, -\theta) \end{cases} \quad f_{d1} = f_{d2} \tag{17}$$

After SAR imaging, imaging results of targets with the same Doppler frequency in the same range bin will be aliased in the same imaging unit. Therefore, it is impossible to distinguish targets with opposite azimuth angles within the same range bin only by single-channel Doppler resolution. From Equation (12), we can see that the steering vector of the target is an odd function of the azimuth angle $\theta$. Therefore, the steering vectors of the ambiguous targets are different.

$$\begin{aligned} \mathbf{V}_1 &= \mathbf{V}(\theta) = \exp\left\{ -j\frac{2\pi}{\lambda} y \sin \alpha \sin \theta \right\} \\ \mathbf{V}_2 &= \mathbf{V}(-\theta) = \exp\left\{ -j\frac{2\pi}{\lambda} y \sin \alpha \sin(-\theta) \right\} \end{aligned} \tag{18}$$

Then the signal model of FLMC-SAR be rewritten as

$$\begin{aligned} s_3(\hat{t}, t_m; r, \theta, \alpha, y) &= s_2(\hat{t}, t_m; r, \theta, \alpha, y) + s_2(\hat{t}, t_m; r, -\theta, \alpha, y) \\ &= \text{sinc}\left\{ B\left[ \hat{t} - \frac{(r - r_0)}{c} \right] \right\} \cdot \exp(-j2\pi f_d t_m) \cdot (A_{P1} \cdot \mathbf{V}_1 + A_{P2} \cdot \mathbf{V}_2). \end{aligned} \tag{19}$$

Equation (19) has realized the range–azimuth decoupling of FLMC-SAR signals. Now, we can process the signals in each of the same range bin.

### 3.3. FLMC-SAR Observation Model

For any imaging unit in an FLMC-SAR image, the array's degree of freedom provides the possibility to solve the Doppler ambiguity. According to Equation (19), the following spatial observation model for each FLMC-SAR imaging unit with the same range bin can be obtained:

$$\mathbf{S}(r,\theta)_{K\times1} = \mathbf{H}_{K\times2}\mathbf{A}(r,\theta)_{2\times1} + \boldsymbol{\nu}_{K\times1} \tag{20}$$

where $\mathbf{S}(r,\theta)_{K\times1}$ is the imaging results for all channel for each FLMC-SAR imaging unit, $\mathbf{A}(r,\theta)_{2\times1}$ is the reflectivity coefficient vector of ambiguous targets, $\boldsymbol{\nu}_{K\times1}$ is the additive measurement noise term, and $\mathbf{H}_{K\times2}$ is the steering vector matrix composed of ambiguous targets.

$$\mathbf{H}_{K\times2} = [h_1, h_2] \tag{21}$$

where $h_i$ is the steering vector of the $i$th target.

$$h_i = \begin{bmatrix} \exp\left(-j\frac{2\pi}{\lambda}d\sin\alpha_i\sin\theta_i\right) \\ \exp\left(-j\frac{2\pi}{\lambda}2d\sin\alpha_i\sin\theta_i\right) \\ ... \\ \exp\left(-j\frac{2\pi}{\lambda}Kd\sin\alpha_i\sin\theta_i\right) \end{bmatrix}_{K\times1} \tag{22}$$

where $\theta_i$ and $\alpha_i$ are the azimuth angle and pitch angle of the target, respectively.

The order of the column vectors in the steering vector matrix $\mathbf{H}_{K\times2}$ corresponds to the order of the reflectivity coefficient of ambiguous targets in $\mathbf{A}(r,\theta)_{2\times1}$. It is worth noting that Doppler ambiguity is two-dimensional in the FLMC-SAR system. Therefore, the steering vector matrix $\mathbf{H}_{K\times2}$ only contains two column vectors. Based on the accuracy of the steering vector, Doppler ambiguity can be solved by spatial methods such as beamforming. However, in practical applications, the inevitable array error between channels causes the space–time characteristic mismatch of the targets and reduces the accuracy of the ambiguous targets steering vector, resulting in the degradation of the Doppler ambiguity resolving performance. To correct the array errors, the following observation model of FLMC-SAR is given in this paper.

$$\mathbf{S}(r,\theta)_{K\times1} = \boldsymbol{\varpi}_{K\times K}\mathbf{H}_{K\times2}\mathbf{A}(r,\theta)_{2\times1} + \boldsymbol{\nu}_{K\times1} \tag{23}$$

The array errors of each channel are expressed as follows:

$$\boldsymbol{\varpi}_{K\times K} = \begin{bmatrix} \rho_1 & & & & & \\ & \rho_2 & & & & \\ & & \ddots & & & \\ & & & \rho_k & & \\ & & & & \ddots & \\ & & & & & \rho_K \end{bmatrix} \tag{24}$$

where $\rho_k$ denotes the steering vector of the $k$th channel.

## 4. Sparsity-Based Array Error Estimation and Doppler Ambiguity Resolving

### 4.1. Improved Quasi-Newton Kernel

Following the Bayesian principle, the cost function of the maximum a posteriori (MAP) estimator [46–49] for the constrained optimization problem of array error estimation can be obtained.

$$\arg\min\ J(\mathbf{A}(r,\theta), \boldsymbol{\varpi}) \tag{25}$$

$$J(\mathbf{A}(r,\theta), \boldsymbol{\varpi}) = \|\mathbf{S}(r,\theta) - \boldsymbol{\varpi}\mathbf{H}\mathbf{A}(r,\theta)\|_F^2 + \omega\|\mathbf{A}(r,\theta)\|_1 \tag{26}$$

where $\omega$ is the regularization parameter related to the sparsity and noise level of the signal. The cost function consists of two terms. The first term in Equation (25) is the data fidelity term of the estimator. The second term in Equation (25) incorporate the signal sparsity, which balances the reconstruction accuracy with the sparsity of the obtained solutions. As Doppler ambiguity is two-dimensional in the FLMC-SAR imaging, the results that need to be reconstructed in the observation model of Equation (23) are at most 2-sparse.

In this paper, an improved quasi-Newton method is proposed to solve the above constrained optimization problem, which uses a 2-step iterative processing. There are two steps in each iteration: image reconstruction and array error estimation Algorithm 1. The iterative process of the proposed algorithm is shown as follows.

---

**Algorithm 1:** Improved quasi-Newton

> **Range cycle:** Traverse all range bin
>> **Azimuth cycle:** Traverse all $N$ azimuth bin within the same range bin
>>> **Input:** The signal of an imaging unit within the same range unit $\mathbf{S}(r, \theta)$, the steering vector matrix of the imaging unit $\mathbf{A}(r, \theta)$, $k=0$.
>>> **Step 1 Image reconstruction:**
>>>
>>> $$A^{k+1}(r, \theta) = \underset{A(r,\theta)}{\arg\min} \; J\left(\mathbf{A}^k(r, \theta), \mathbf{æ}^k\right) \tag{27}$$
>>>
>> **end**
>> **Step 2 Array error estimation:**
>>
>> $$\mathbf{æ}^{k+1} = \underset{\mathbf{æ}}{\arg\min} \sum_{\theta=\theta_1}^{\theta_N} J\left(\mathbf{A}^{k+1}(r, \theta), \mathbf{æ}^k\right) \tag{28}$$
>>
>> $k = k + 1$ until
>>
>> $$\sum_{\theta=\theta_1}^{\theta_N} \|\mathbf{S}(r, \theta) - \mathbf{æHA}(r, \theta)\|_F^2 \leq \zeta \tag{29}$$
>>
> **end**

---

The initialization array error matrix $\mathbf{æ}_0$ is set to be the identity matrix. In order to reduce the complexity of the proposed method, the array error is estimated by combining all imaging units within the same distance unit, which also reduces the influence of singularities and enhances the robustness of the algorithm.

Step 1: To achieve image reconstruction in each iteration, we first need to solve the constrained optimization problem of Equation (27). To avoid the nondifferentiability of the $\ell_1$ norm in Equation (27), we use the following smooth approximation:

$$\|\mathbf{A}(r, \theta)\|_1 = \sum_{\theta=\theta_1}^{\theta_2} \left(a^2(r, \theta) + \xi\right)^{\frac{1}{2}} \tag{30}$$

where $\xi$ is a small nonnegative constant, and $a(r, \theta)$ is the element in $\mathbf{A}(r, \theta)$.

The cost function is slightly modified as

$$J(\mathbf{A}(r, \theta), \mathbf{æ}) = \|\mathbf{S}(r, \theta) - \mathbf{æHA}(r, \theta)\|_F^2 + \omega \sum_{\theta=\theta_1}^{\theta_2} \left(a^2(r, \theta) + \xi\right)^{\frac{1}{2}}. \tag{31}$$

Then the conjugate gradient of cost function in Equation (31) over $\mathbf{A}(r, \theta)$ can be obtained:

$$\nabla J_{\mathbf{A}(r,\theta)} = 2(\mathbf{æH})^H \mathbf{æHA}(r, \theta) + \omega \mathbf{U}_{\mathbf{A}(r,\theta)} \mathbf{A}(r, \theta) - 2\mathbf{æH}^H \mathbf{S}(r, \theta) \tag{32}$$

where $\mathbf{U_A}$ is a diagonal matrix, represented as follows:

$$\mathbf{U}_{\mathbf{A}(r,\theta)} = \begin{bmatrix} \frac{1}{\sqrt{|a(r,\theta_1)|^2+\xi}} & 0 \\ 0 & \frac{1}{\sqrt{|a(r,\theta_2)|^2+\xi}}. \end{bmatrix} \tag{33}$$

Therefore, the Hessian matrix can be approximated as follows:

$$\mathbf{H}_{\mathbf{A}(r,\theta)} = 2(\text{æH})^H \text{æH} + \omega \mathbf{U}_{\mathbf{A}(r,\theta)} \tag{34}$$

In this way, we can obtain the iteration solver of image reconstruction.

$$\mathbf{A}^{k+1}(r,\theta) = \mathbf{A}^k(r,\theta) - \left[\mathbf{H}_{\mathbf{A}^k(r,\theta)}\right]^{-1} \nabla J_{\mathbf{A}(r,\theta)} \tag{35}$$

Step 2: Array error estimation is achieved by solving the constrained optimization problem of Equation (28).

The conjugate gradient of Equation (31) over **æ** can be obtained:

$$\nabla J_{\text{æ}} = \sum_{\theta=\theta_1}^{\theta_N} \left[ 2\mathbf{A}(r,\theta)^H \mathbf{H}^H \text{æH} \mathbf{A}(r,\theta) - 2\mathbf{A}(r,\theta)^H \mathbf{H}^H \mathbf{S}(r,\theta) \right]. \tag{36}$$

Let $\nabla J_{\text{æ}} = 0$, then we can obtain the iteration solver of array error estimation; **æ** is a diagonal matrix, so the update rule can be expressed as

$$\text{æ}^{k+1} = \text{diag}[\rho_a \exp(j\rho_b)] \tag{37}$$

where $\rho_a$ and $\rho_b$ is the amplitude error and phase error of the array, respectively.

$$\rho_a = \sum_{\theta=\theta_1}^{\theta_N} \left\{ \text{abs}[\mathbf{S}(r,\theta)]./\text{abs}(\mathbf{HA}(r,\theta)) \right\} \tag{38}$$

$$\rho_b = \sum_{\theta=\theta_1}^{\theta_N} \left\{ \text{angle}\{\mathbf{S}(r,\theta) \odot \text{conj}[\mathbf{HA}(r,\theta)]\} \right\} \tag{39}$$

where $\odot$ is the Hadamard product. Each range cycle continues until the Equation (29) is satisfied, then **æ** and **A** are, respectively, the reconstructed image and the estimation of array errors within this range bin.

### 4.2. Computational Complexity

The above operations are performed on all range bins to achieve full-area FLMC-SAR ambiguity resolving and imaging with array error correction. The flowchart of the proposed method is shown in Figure 3. The proposed method mainly consists of two steps: FLMC-SAR imaging and joint Doppler ambiguity resolving and array error correction.

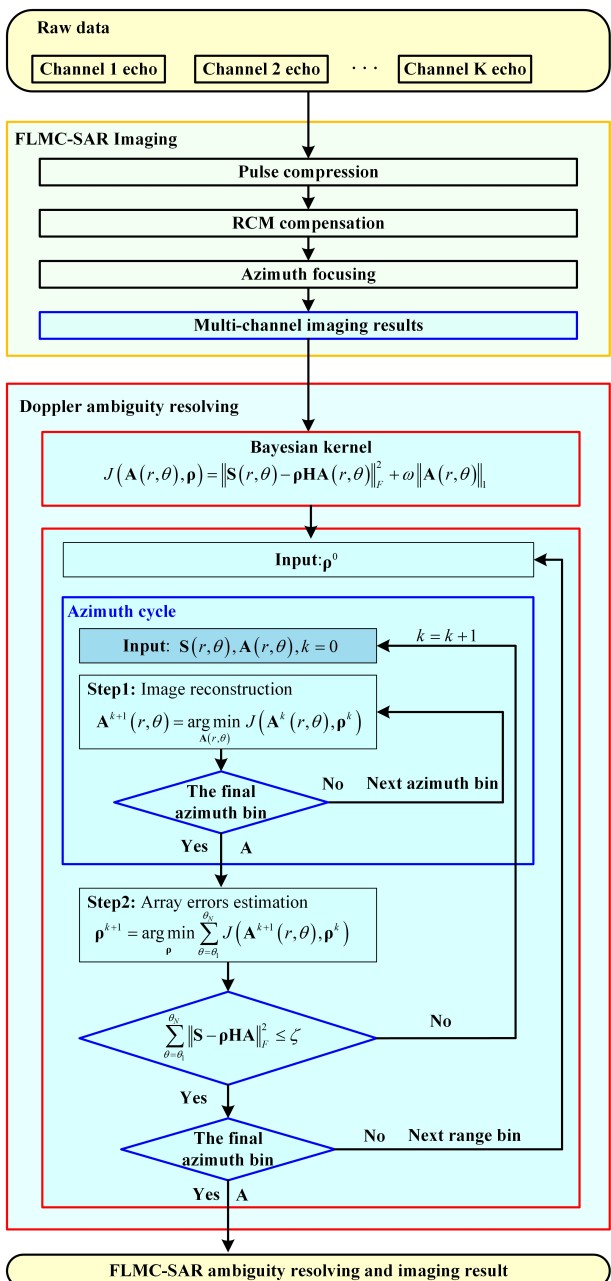

**Figure 3.** Flowchart of the proposed method.

Assume that the number of channels is *K*, the number of azimuth bins is *N*, and the the number of range bins is *M*. Then the computational complexity of the proposed method is analyzed as follows.

(1) FLMC-SAR imaging contains range pulse compression, RCMC, and azimuth focusing for multi-channels. Range pulse compression needs $KN$ times fast Fourier transform (FFT) for the vector with the size of $M \times 1$, and $KN$ times inverse fast Fourier transform (IFFT) for the vector with the size of $M \times 1$. RCMC needs $KN$ times FFT for the vector with the size of $M \times 1$, $KN$ times IFFT for the vector with the size of $M \times 1$, $KM$ times FFT for the vector with the size of $N \times 1$, and $KM$ times IFFT for the vector with the size of $N \times 1$. Azimuth focusing needs $KM$ times FFT for the vector with the size of $N \times 1$. The total computation of FLMC-SAR imaging is $O(KNM(\log N + \log M))$.

(2) Joint Doppler ambiguity resolving and array error correction. Suppose the number of iterations is $N_{it}$. For each range bin, imaging reconstruction needs $M$ times sparse reconstruction, which needs $N_{it}$ times matrix inversion for the matrix with the size of $K \times K$.

Array error estimation needs $2N$ times the Hadamard product for the diagonal matrix with the size of $K \times K$. The total computation of joint Doppler ambiguity resolving and array error correction is $O(N_{it}NMK^3 + 2NK)$.

Therefore, the total computation of the proposed method is of the order $O(NM(K \log N + K \log M + N_{it}K^3))$

## 5. Results

### 5.1. Point Target Simulation

We first give the point target simulation experiment to verify the efficacy of the proposed array error estimation and imaging reconstruction method. As shown in Figure 4, there are nine point targets in the original reference image. The red trajectory extension line in Figure 4 divides the image into the left area and the right area.

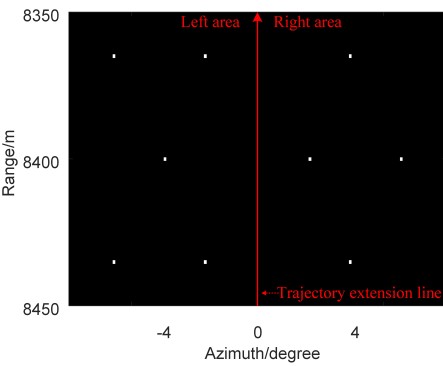

**Figure 4.** Original reference image.

Five point targets and four point targets are set in the left area and the right area, respectively. The coordinates of the point targets are shown in Table 1. Table 2 shows the simulation parameters. The SNR is set to 20 dB. The array error is also added to the echo.

**Table 1.** Point target coordinates.

| Left Area | | | | | Right Area | | | |
|---|---|---|---|---|---|---|---|---|
| $P_1$ | $P_2$ | $P_3$ | $P_4$ | $P_5$ | $P_6$ | $P_7$ | $P_8$ | $P_9$ |
| $(Rs_1, -5)$ | $(Rs_1, -3)$ | $(Rs_2, -4)$ | $(Rs_3, -5)$ | $(Rs_3, -3)$ | $(Rs_1, 4)$ | $(Rs_2, 3)$ | $(Rs_2, 5)$ | $(Rs_3, 4)$ |

The point targets' imaging results are shown in Figure 5. Figure 5a–d indicate the ambiguous imaging results, the ambiguity resolving results of beamforming without the array calibration, the imaging result of the proposed method, and the ambiguity resolving results of beamforming with the array calibration, respectively.

**Table 2.** Simulation parameters.

| | | | |
|---|---|---|---|
| Carrier frequency | 30 GHz | Platform height | 4000 m |
| Bandwidth | 55 MHz | Platform velocity | 84 m/s |
| Number of array element | 9 | Reference slant range | 8400 m |
| PRF | 2500 Hz | Synthetic aperture time | 0.82 s |

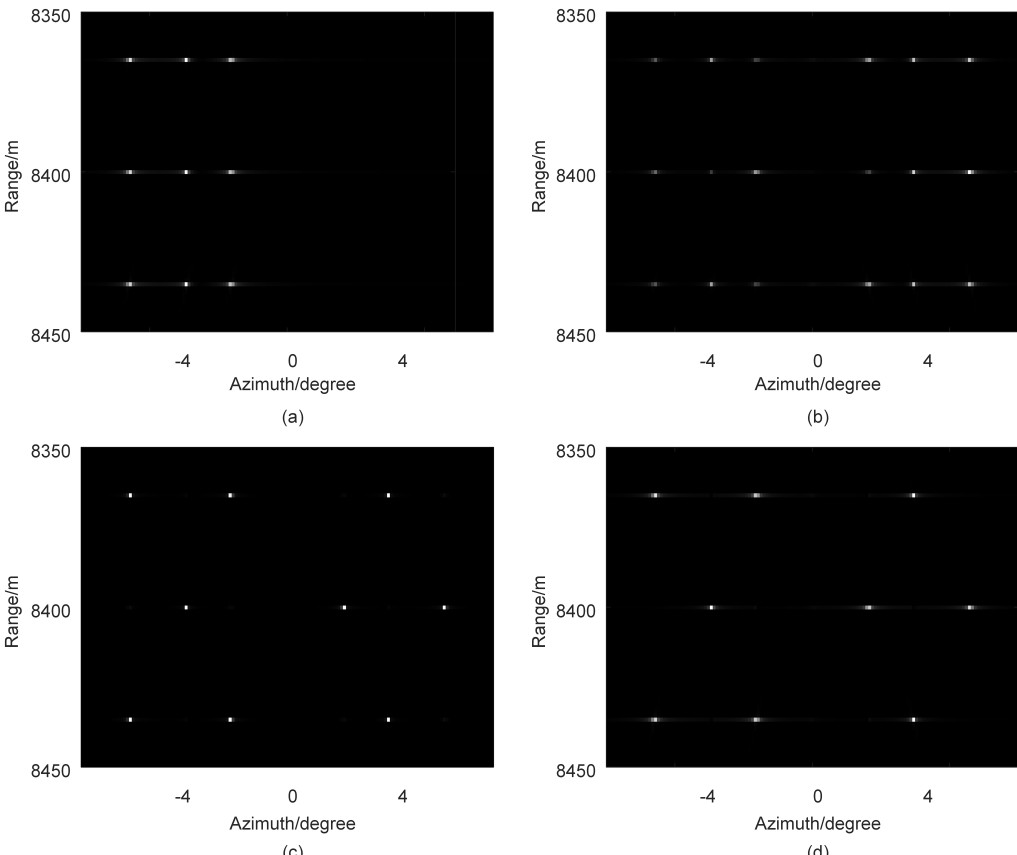

**Figure 5.** Imaging results of point targets: (**a**) ambiguous imaging results; (**b**) ambiguity resolving results of beamforming without the array calibration; (**c**) imaging result of the proposed method; (**d**) ambiguity resolving results of beamforming with the array calibration.

### 5.2. Surface Target Simulation

To verify the effectiveness of the proposed method for natural scenes, a surface target simulation is presented in this section. The original reference image is shown in Figure 6. The illuminated area in Figure 6 is divided into the left area and the right area by the red trajectory extension line. Table 2 shows the simulation parameters. The SNR is set to 20 dB. The array error is also added to the echo.

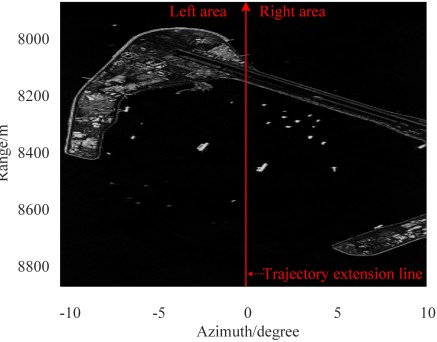

**Figure 6.** Original reference image.

The image results of the surface target are shown in Figure 7. Figure 7a–d indicate the ambiguous imaging results, the ambiguity resolving results of beamforming without the array calibration, the imaging result of the proposed method, and the ambiguity resolving results of beamforming with the array calibration, respectively.

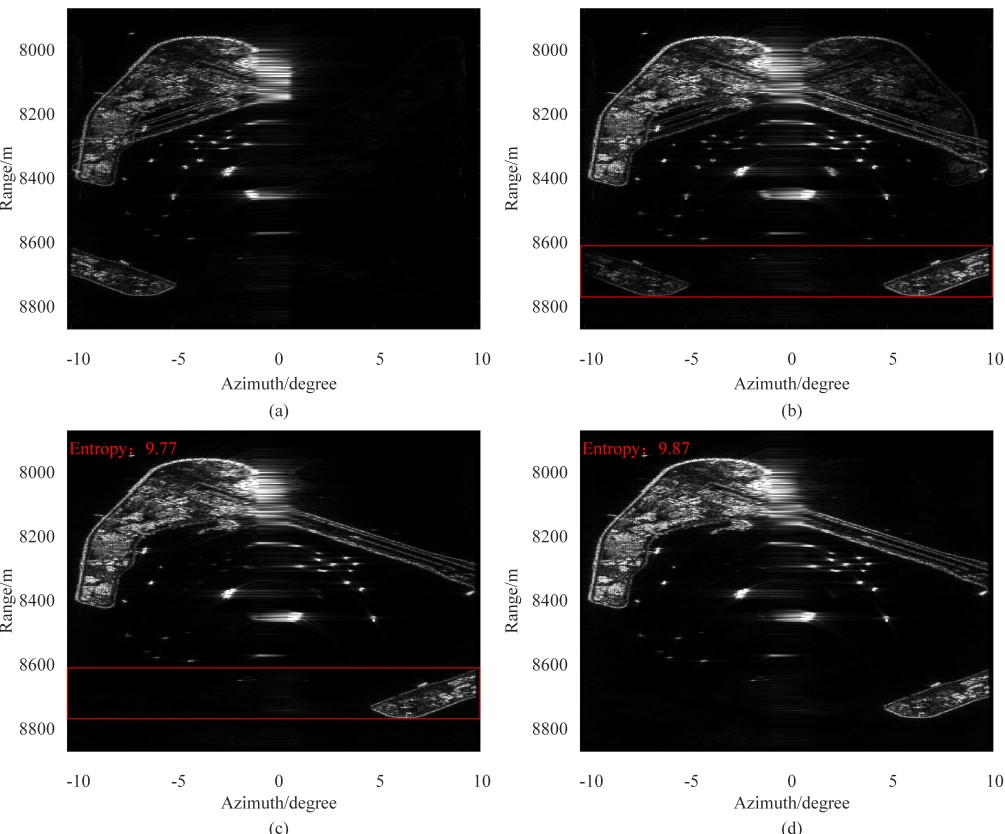

**Figure 7.** Imaging results of the surface target: (**a**) ambiguous imaging results; (**b**) ambiguity resolving results of beamforming without the array calibration; (**c**) imaging result of the proposed method; (**d**) ambiguity resolving results of beamforming with the array calibration.

*5.3. Real Data Experiment*

To demonstrate the efficacy of the proposed method in a practical application, we carried out a real-data experiment in this section. The real-data experiment was performed by a K-band FLMC-SAR. The radar is equipped with a five-channel array antenna. The radar platform is mounted on the aircraft at an altitude of 4000 m and a speed of 80 m/s. The relevant experiment parameters are shown in Table 3.

**Table 3.** Experiment parameters.

| | | | |
|---|---|---|---|
| Carrier frequency | 30 GHz | Platform height | 4000 m |
| Bandwidth | 55 MHz | Platform velocity | 80 m/s |
| Number of array element | 9 | Reference slant range | 8000 m |
| PRF | 6000 Hz | Synthetic aperture time | 1.3 s |

Imaging results of the real-data experiment are shown in Figure 8. Figure 8a–c indicate the ambiguous imaging results, the ambiguity resolving results of beamforming, and the imaging result of the proposed method. The satellite image of the imaging area is shown in Figure 9.

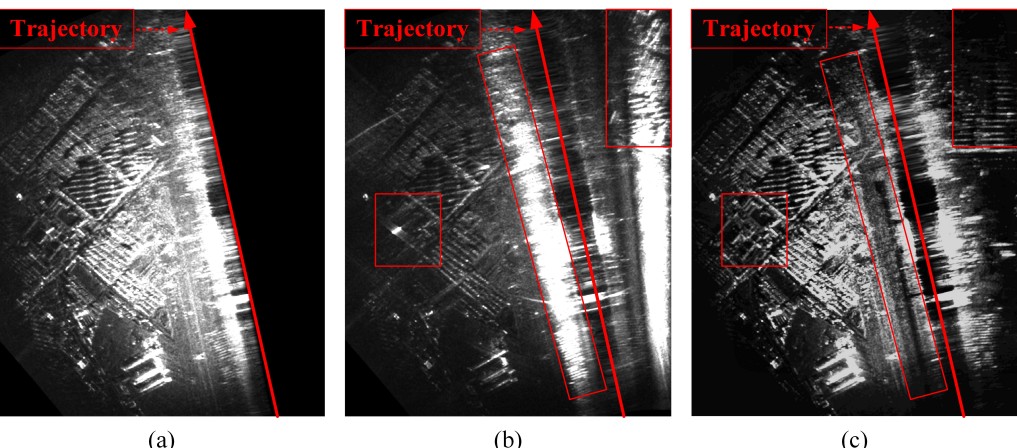

**Figure 8.** Imaging results of the real-data experiment: (**a**) ambiguous imaging results; (**b**) ambiguity resolving results of beamforming; (**c**) imaging result of the proposed method.

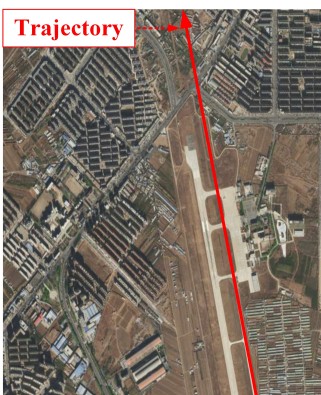

**Figure 9.** Satellite image of the imaging area.

## 6. Discussion

In point target simulation, Figure 5a shows that the targets in the left area and right area are aliased, leading to the left and right Doppler ambiguity in FLMC-SAR. In order to solve the Doppler ambiguity, a beamforming-based left–right Doppler ambiguity resolving is proposed [23]. However, due to the existence of array errors, beamforming cannot effectively solve the left–right Doppler ambiguity without array correction, as shown in Figure 5b. In order to enhance the robustness of Doppler ambiguity resolving for FLMC-SAR, we propose a sparsity-based array calibration method, which can be used for Doppler ambiguity resolving for FLMC-SAR. The azimuth ambiguity-to-signal ratio (AASR) of the nine point targets is shown in Table 4.

**Table 4.** AASR of point targets.

|  | $P_1$ | $P_6$ | $P_2$ | $P_8$ | $P_3$ | $P_7$ | $P_4$ | $P_9$ | $P_5$ |
|---|---|---|---|---|---|---|---|---|---|
| Beamforming | 3.23 | 6.87 | 2.95 | 9.69 | 2.23 | 3.22 | 3.23 | 6.88 | 2.95 |
| Proposed method | 23.42 | 24.79 | 25.03 | 23.72 | 24.71 | 25.11 | 23.77 | 23.56 | 25.41 |

After array calibration, the AASR of all targets is greater than 23 dB. It is concluded from the practical application that an AASR greater than 20 dB is convenient for people or computers to identify the target. Therefore, it can be seen that the proposed method is

necessary for the FLMC-SAR system. To evaluate the accuracy of array error estimation of the proposed method, we define the mean square error of the array as follows:

$$\text{MSE} = \|\boldsymbol{æ} - \hat{\boldsymbol{æ}}\|_F^2 \tag{40}$$

where $\hat{\boldsymbol{æ}}$ is the array error estimation. In order to reduce the complexity of the method, the array error estimation is based on the range–azimuth decoupling. Figure 10 shows the MSE of the array errors in the three range bins ($Rs_1$, $Rs_2$, and $Rs_3$ where targets are located) after each iteration.

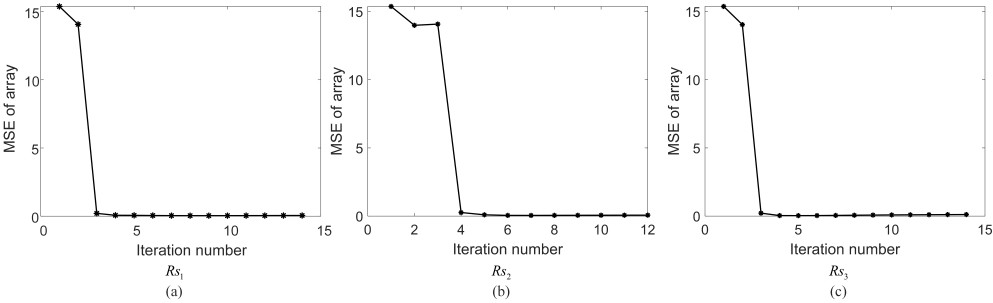

**Figure 10.** MSE of the array errors: (**a**) $Rs_1$ range bin; (**b**) $Rs_2$ range bin; (**c**) $Rs_3$ range bin.

The proposed method converges within ten iterations. Finally, the estimation of the array errors in the range bins $Rs_1$, $Rs_2$, and $Rs_3$ are shown in Figure 11. Compared with the actual array errors added into the echo, the estimation of array errors in each range bin are consistent with the array errors, which verifies the effectiveness of the array error estimation. Furthermore, the MSE of the array phase errors is less than $\pi/8$. After array calibration, the influence of the array errors on the Doppler ambiguity resolving can be ignored.

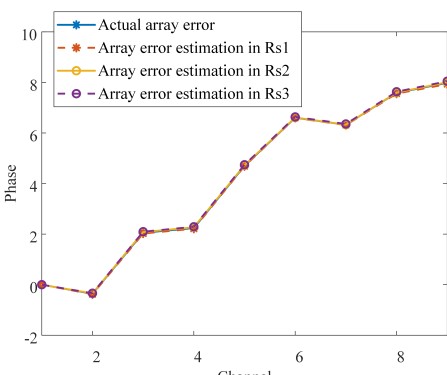

**Figure 11.** Estimation of the array errors.

We use the array error estimation from the proposed method for array calibration. The Doppler ambiguity resolving results via beamforming are shown in Figure 5d. Doppler ambiguity can be resolved well. The proposed method not only performs array error estimation but also achieves sparse image reconstruction. The azimuth pulse response functions of beamforming and the proposed method are shown in Figure 12. Compared with the beamforming-based Doppler ambiguity resolving, the proposed method introduces data fidelity into the image reconstruction model and combines signal sparsity, thus achieving sidelobe reduction and noise suppression.

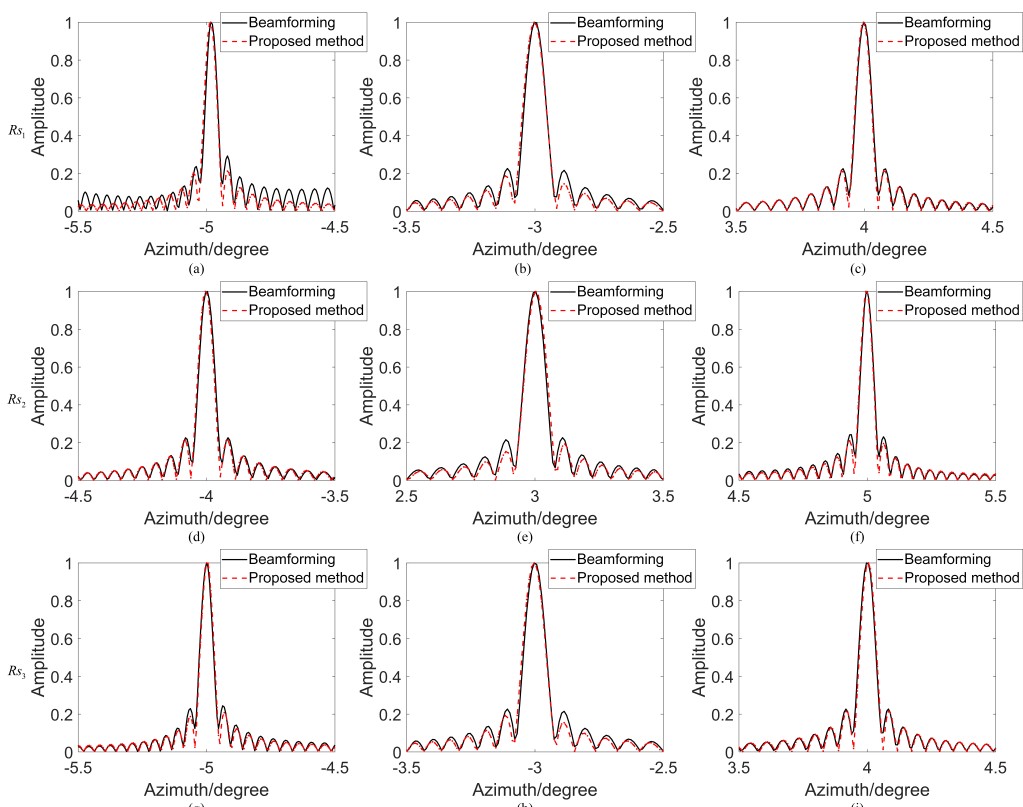

**Figure 12.** Azimuth pulse response functions: (**a**) $P_1$; (**b**) $P_2$; (**c**) $P_3$; (**d**) $P_4$; (**e**) $P_5$; (**f**) $P_6$; (**g**) $P_7$; (**h**) $P_8$; (**i**) $P_9$.

Peak sidelobe ratio (PSLR), integrated sidelobe ratio (ISLR), and impulse response width (IRW) of the point impulse responses are shown in Table 5. The values of IRWs obtained by the proposed method are basically consistent with those obtained by beamforming. However, the values of PSLR and ISLR obtained by the proposed method are significantly lower than those obtained by beamforming, which verifies that the proposed method has certain sidelobe suppression and noise suppression capabilities. Moreover, we also varied the AASR of all point targets under different SNRs, as shown in Figure 13. Thus, if SNR is greater than 10 dB, AASR is greater than 20 dB. The array error estimation and Doppler ambiguity resolving are based on the image domain. The range pulse compression and azimuth focusing can significantly improve the SNR so that the condition of SNR being greater than 10 dB can be easily met in practical applications.

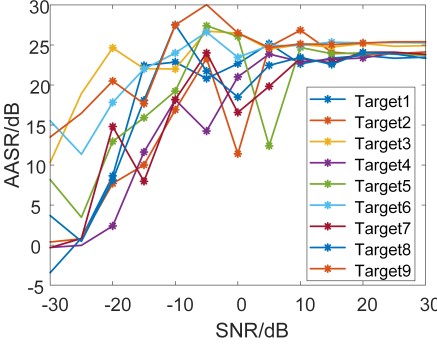

**Figure 13.** AASR of the point targets under different SNRs.

In surface target simulation, left–right Doppler ambiguity makes the image results of the left scene and that of the right aliased together, as shown in Figure 7a. The imaging

results of beamforming-based Doppler ambiguity resolving are shown in Figure 7b. Due to the array errors, the performance of beamforming deteriorates. The average AASR of the targets in the red rectangle in Figure 7b is 6.79 dB. Clearly ambiguous residual images can be seen in Figure 7b. The proposed method was used for array error estimation and imaging reconstruction, and the image results are shown in Figure 7c. The average AASR of the targets in the red rectangle in Figure 7c is 21.81 dB. The performance of Doppler ambiguity resolving has been obviously improved.

**Table 5.** PSLR, ISLR, and IRW obtained by beamforming and the proposed method.

| Azimuth Angle/° | Reference Slant Range | Beamforming | | | Proposed Method | | |
|---|---|---|---|---|---|---|---|
| | | PSLR/dB | ISLR/dB | IRW/m | PSLR/m | ISLR/m | IRW/m |
| −5 | $Rs_1$ | −12.19 | −1.03 | 6.05 | −13.46 | −10.27 | 6.05 |
| 4 | $Rs_1$ | −12.94 | −1.60 | 7.43 | −13.39 | −1.96 | 7.30 |
| −1 | $Rs_1$ | −12.90 | −1.75 | 10.03 | −14.53 | −12.24 | 10.03 |
| 5 | $Rs_2$ | −12.30 | −1.06 | 5.89 | −13.54 | −10.53 | 5.89 |
| −1 | $Rs_2$ | −12.91 | −1.59 | 7.41 | −13.38 | −1.94 | 7.40 |
| 3 | $Rs_2$ | −12.89 | −1.70 | 9.61 | −14.30 | −11.88 | 9.95 |
| −1 | $Rs_3$ | −12.24 | −1.09 | 5.95 | −13.65 | −10.79 | 5.95 |
| 4 | $Rs_3$ | −12.88 | −1.58 | 7.43 | −13.38 | −1.92 | 7.30 |
| −1 | $Rs_3$ | −12.89 | −1.69 | 10.02 | −14.29 | −11.82 | 10.02 |

To evaluate the accuracy of array error estimation of the proposed method, we also give the MSE of the array errors in the three range bins ($Rs_1$, $Rs_2$, and $Rs_3$) after each iteration, as shown in Figure 14. The proposed method converges within ten iterations. Finally, the estimation of array errors in the three range bins ($Rs_1$, $Rs_2$, and $Rs_3$) are shown in Figure 15. Compared with the actual array errors added into the echo, the estimation of array errors in the three range bins are consistent with the array errors, which verifies the effectiveness of the array error estimation. The MSE of the array phase errors is less than $\pi/8$. On this basis, the Doppler ambiguity can be resolved by beamforming to get unambiguous imaging results, as shown in Figure 7d. Comparing Figure 7c,d, the imaging result of the proposed method has a lower entropy than the ambiguity resolving results of beamforming with the array calibration, which verifies that the proposed method has sidelobe suppression and noise suppression capabilities.

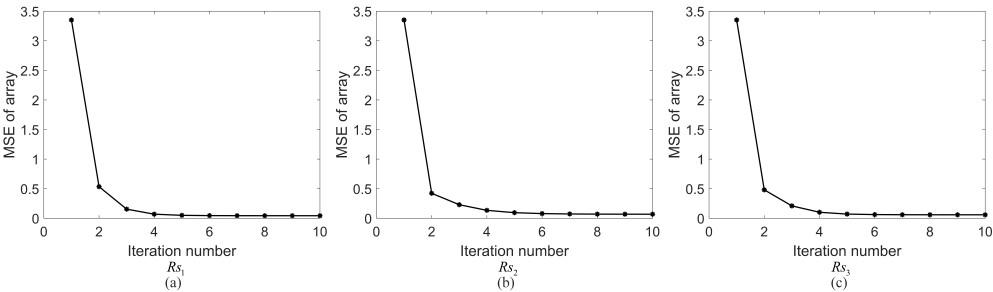

**Figure 14.** MSE of the array errors: (**a**) $Rs_1$ range bin; (**b**) $Rs_2$ range bin; (**c**) $Rs_3$ range bin.

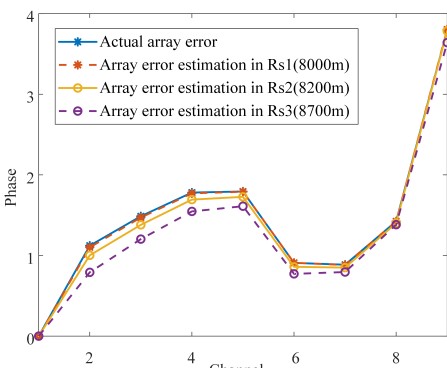

**Figure 15.** Estimation of the array errors.

To verify the robustness of the method under noise, we also give the image results of the surface target under different SNRs, as shown in Figure 16. Figure 16a–e are the imaging results of the surface target under SNRs of −10 dB, −10 dB, 0 dB, 10 dB, and 20 dB, respectively. Calculating the average AASR of the image results obtained by the beamforming and the proposed method, the curve of AASR changing with SNR is shown in Figure 16f. It can see that the average AASR of the image results obtained by the proposed method is more than 20 dB. The AASR improves by more than 10 dB compared to the image results obtained by the beamforming.

In the real-data experiment, the targets of the left area are aliased with that of the right area, and the imaging results are ambiguous. Based on Figure 8a, beamforming is used to solve the Doppler ambiguity, and the image results are shown in Figure 8b. One can see that the Doppler ambiguity has been resolved in some areas. However, due to the effect of array errors, most areas are still seriously ambiguous, such as the area highlighted by the red matrix. Due to Doppler ambiguity, the airstrip in Figure 16 cannot be distinguished from the ambiguous targets in Figure 8a. After the beamforming-based Doppler ambiguity resolving, due to the array error, the airstrip is still indistinct in Figure 8b. The proposed method is used to realize array error estimation and imaging reconstruction, and the airstrip is clearly distinguished, as shown in Figure 8c. Experimental results verify that the proposed method can be applied to FLMC-SAR system to realize unambiguous forward-looking imaging.

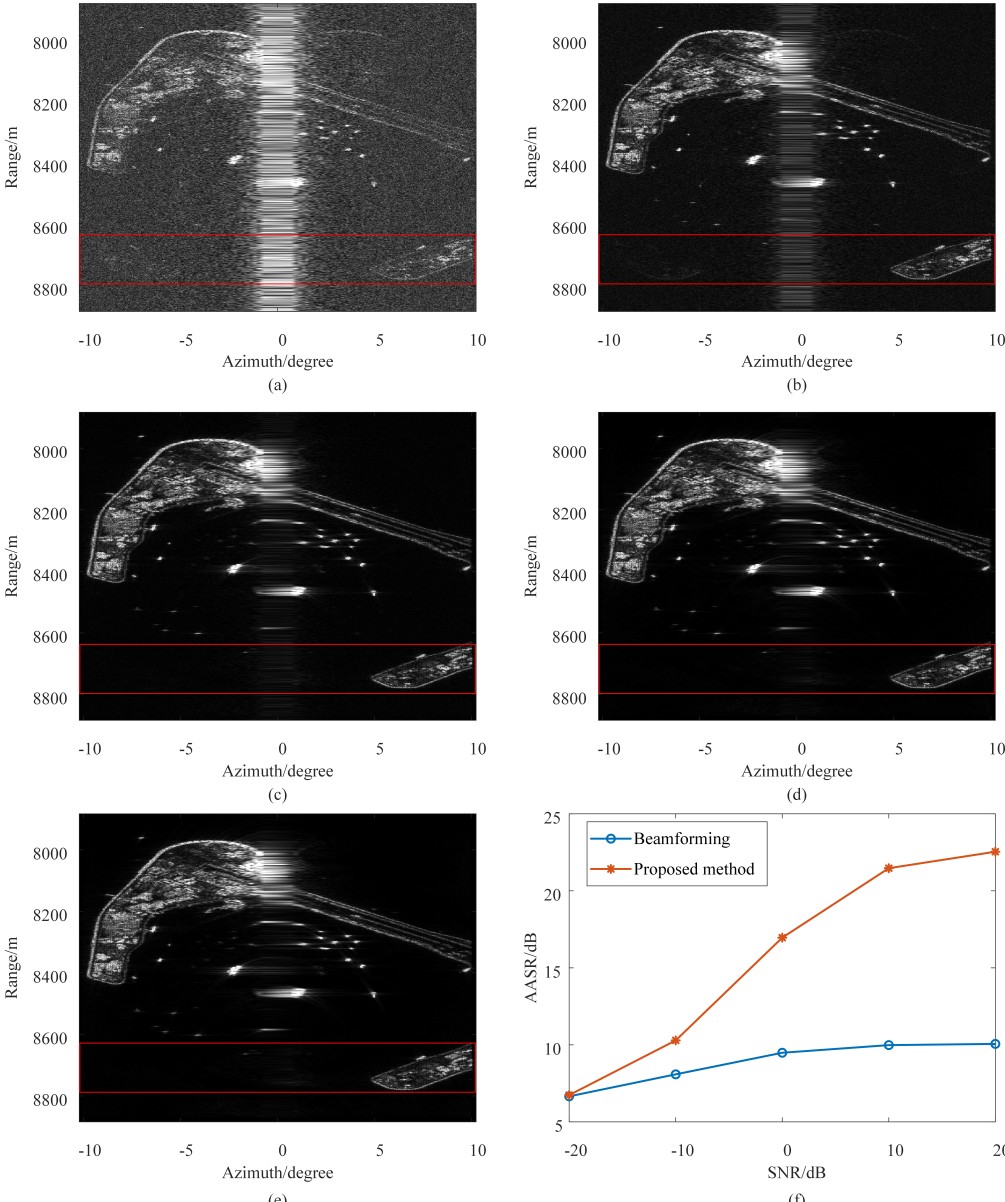

**Figure 16.** Imaging results of surface target under different SNRs: (**a**) imaging results under the SNR of −10 dB; (**b**) imaging results under the SNR of −10 dB; (**c**) imaging results under the SNR of 0 dB; (**d**) imaging results under the SNR of 10 dB; (**e**) imaging results under the SNR of 20 dB; (**f**) AASR of the imaging results under different SNRs.

## 7. Conclusions

For the FLMC-SAR system, the array error is an important factor that causes performance degradation of beamforming-based Doppler ambiguity resolving. In this paper, a sparsity-based array calibration and ambiguity resolving method is proposed for enhancing the robustness of FLMC-SAR imagery. First, the observation model of FLMC-SAR Doppler ambiguity combined with array error is derived. The model shows that array errors will lead to the mismatch of space–time characteristics of the targets, causing performance degradation of Doppler ambiguity resolving. Based on observational models, a constrained optimization problem for FLMC-SAR imaging is formulated, transforming the Doppler ambiguity resolving and array error estimation into a sparse recovery problem. Then a modified quasi-Newton method is proposed to realize both array error estimation and Doppler ambiguity resolving. Finally, simulation and real-data experiments verify the effectiveness of the proposed method.

**Author Contributions:** Conceptualization, J.L. and L.Z.; methodology, L.Z.; software, J.L.; validation, X.W.; investigation, X.W.; resources, X.W.; data curation, Y.C.; writing—original draft preparation, J.L.; writing—review and editing, Y.C. All authors have read and agreed to the published version of the manuscript.

**Funding:** This research received no external funding.

**Institutional Review Board Statement:** Not applicable.

**Informed Consent Statement:** Not applicable.

**Data Availability Statement:** Not applicable.

**Acknowledgments:** The authors would like to thank the anonymous reviewers for their valuable comments to improve the quality of this article.

**Conflicts of Interest:** The authors declare no conflict of interest.

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
