# Peer review of "Sparsity-Based Joint Array Calibration and Ambiguity Resolving for Forward-Looking Multi-Channel SAR Imagery"

_remotesensing, doi:10.3390/rs15030647_

Round 1
Reviewer 1 Report
The paper addresses the several issues for forward-looking SAR imaging and suggests contribution to resolve challenging problems.
One of the missing issues is how to design the multi channel structures. The performance of the proposed algorithm may be dependent on the number of channels, beamwidths, spacings between the antennas. The complexity and reliability can also be highly dependent on the size of the K. These issues are not discussed in the paper.
The parameters for simulation and real experiment are different in Table 2 and 3. Is there any reason for this? For the same channel number of 9, PRFs are given as 2500 and 6000 respectively, while their Iteration performances appear very similar. I assume the convergence speed should be much slower in real target scene having high clutter levels. Comments should be addressed.
I doubt we need multiple number of K to recover Doppler ambiguity instead of 2 or 3 channels.
In table 5, it is claimed that the ISLR, PSLR performances have significantly improved while resolutions are kept the same. I wonder how this could be achieved, when the same power levels are employed in both cases.
Th
The point
Reviewer 2 Report
The article presents a method for antenna array calibrating. The developed method is well described, and there are no general issues according to the clarity of the presentation. However, some aspects of the article or process must be considered or clarified.
(eq. 7) The model of downconverted signal seems incorrect according to the transmitted signal model. No direct connection between eq 6 and 7 can be found.
(line 124) What kind of pulse compression is used? Is it match filtering or FFT after mixing with the transmitted one?
(eq. 8) This signal model seems too simple to describe the phenomenon of range-angle-Doppler dependency of reflectivity. Please discuss this a litter bit more.
(eq. 30) This equality has to be proven. Where it comes from? We have the L1 norm on the left side, while on the right side, it looks like the L2 norm.
(eq. 34) Where this Hessian matrix comes from?
Reviewer 3 Report
The Doppler ambiguity is an important problem that forward-looking SAR faced. In this paper, a sparsity-based joint array calibration and ambiguity resolving for forward-looking SAR is proposed to enhance the robustness of FLMC-SAR Imagery. This research work is meaningful for forward-looking SAR, especially the monostatic forward-looking SAR imaging. There have some problems that need to be solved.
(1) In this paper, the proposed method can simultaneously realize both array error estimation and Doppler ambiguity resolving. The details regarding how the method resolves Doppler ambiguity should be analyzed.
(2) Doppler ambiguity makes symmetric targets aliased in the same imaging unit and also within one beam width. How to distinguish all ambiguous targets within one beam width by spatial method?
(3) On line 150, “his Doppler frequency” should be “this Doppler frequency”.
(4) Please check full manuscript for spelling and usage of English.
(5) What is the processing time of simulation and real data experiment? Can the proposed method be used in real-time processors?
(6) The presentation of organizational structure of this article needs to be carefully checked.
Reviewer 4 Report
This manuscript focuses on monostatic forward-looking SAR imagery and presents a sparsity-based joint array calibration and ambiguity resolving method to enhance the robustness of FLMC-SAR Imagery. The space-time characteristic of targets is firstly analyzed, based on which the observation model of FLMC-SAR Doppler ambiguity combined with array error is derived. Then a modified Quasi-Newton method is proposed to realize the array error estimation and Doppler ambiguity resolving via solving the sparse recovery problem. The research work is meaningful and has some interest and contribution to SAR research field, but there are several issues that should be considered.
(1) Whether bistatic forward-looking SAR has the left-right Doppler ambiguity? With proper geometry configurations, is it possible to avoid the left-right Doppler ambiguity for FLMC-SAR imaging
(2) Fig. 12 should give the targets information corresponding to each azimuth pulse response function.
(3) The format of Table 5 should be improved.
(4) In this paper, array errors have a great influence on beamforming for Doppler ambiguity resolving. It is suggested to analyze the influence of array errors of the channel on beamforming.
(5) English expressions need to be checked.
Round 2
Reviewer 1 Report
The issues raised in the first draft have been well addressed.
Simulations process are well organized and neatly applied for the real target scene.
It is understood that Fig. 7 has a good sparity level and the proposed method is well applicable. However, the real target scene exhibits low sparity level and poses a severe difficulty in recovering and resolving the ambiguous targets. Maybe there could be an alternatively simple way of resolving the ambiguities and authors are encouraged to address this issue at the end of the paper.